# Pharmacogenetics of Metformin Transporters Suggests No Association with Therapeutic Inefficacy among Diabetes Type 2 Mexican Patients

**DOI:** 10.3390/ph15070774

**Published:** 2022-06-22

**Authors:** Adiel Ortega-Ayala, Nidia Samara Rodríguez-Rivera, Fernando de Andrés, Adrián LLerena, Eliseo Pérez-Silva, Adriana Guadalupe Espinosa-Sánchez, Juan Arcadio Molina-Guarneros

**Affiliations:** 1Programa de Maestría y Doctorado en Ciencias Médicas, Odontológicas y de la Salud, UNAM, Mexico City 14080, Mexico; ad.orteg@hotmail.com (A.O.-A.); adriana.espinosa9212@gmail.com (A.G.E.-S.); 2Departamento de Farmacología, Facultad de Medicina, UNAM, Mexico City 14080, Mexico; nidia.rodriguez@comunidad.unam.mx; 3INUBE Instituto Universitario de Investigación Biosanitaria de Extremadura, CICAB Clinical Research Centre, Hospital and Medical School of the Extremadura University, 06002 Badajoz, Spain; fandressegura@gmail.com (F.d.A.); allerena@unex.es (A.L.); 4Servicio de Medicina Interna, Hospital Regional de Alta Especialidad Ixtapaluca, Secretaría de Salud, Ixtapaluca 56530, Mexico; eliseops99@yahoo.com

**Keywords:** pharmacogenetics, diabetes type 2, metformin, sulphonylureas, transporters, therapeutic inefficacy

## Abstract

Mexico has been under official epidemiological alert due to diabetes since 2016. This study presents new information on the frequency and variants of metformin transporters OCT1, OCT2, OCT3, *ABCB1*, and *CYP2C9* variants as well. It also reports the association with HbA1c control on 103 DMT2 patients. They were genotyped through real-time PCR (TaqMan assays) and grouped according to treatment: metformin and metformin + glibenclamide. Metformin plasmatic levels were determined through mass spectrometry. The analysis of HbA1c showed statistical significance across genotypes in polymorphisms rs72552763 (*p* = 0.022), rs622342 (*p* = 0.009), rs1128503 (*p* = 0.021), and rs2032582 (*p* = 0.009) within the monotherapy group. Bivariate analysis found no association between any polymorphism and HbA1c control. Two logistic regression models accounted for two diplotypes in OCT1 and *ABCB1*, including statistically significant covariates. The first model yielded significance in age (*p* = 0.026), treatment period [*p* = 0.001], BMI ≥ 25 kg/m^2^ (*p* = 0.043), and combined therapy (*p* < 0.001). There was no association with *GAT/GAT* of rs72552763 or *A/A* rs622342 in OCT1. The second model yielded significance in age (*p* = 0.017), treatment period (*p* = 0.001), BMI ≥ 25 kg/m^2^ (*p* = 0.042), and combined therapy (*p* < 0.001), finding no association with *C/C* of rs1128503 or *G/G* of rs2032582 in *ABCB1*. Our multinomial logistic regression results may benefit future predictive analyses in diabetic populations.

## 1. Introduction

Diabetes mellitus type 2 (DMT2) is a highly prevalent multifactorial and chronic disease characterized by hyperglycemia, insulin resistance, and a decrease in β-cell levels and insulin secretion [1]. It may also give rise to micro-and macro-vascular and neuropathic complications, including nephropathy, coronary artery disease, stroke, peripheral vascular disease, retinopathy, and neuropathy [2]. Globally, DMT2 is one of the most notable premature mortality risk factors, and it currently represents one of the most frequent causes of mortality globally [3]. Mexico holds the sixth largest global DMT2 population, amounting to 8.7 million people. It is estimated that by 2035, this will rise to fifth place with a total of 15.7 million (20–79 years of age) [4]. According to the International Diabetes Federation (IDF), the adjusted prevalence was 13.1% between the 20–79-year-old group in 2017 [5]. On 14 November 2016, the Mexican Secretary of Health declared diabetes and obesity a national epidemiological emergency, a status that prevails today [6]. The Mexican Official Norm NOM-015-SSA2-1994 for diabetes prevention, treatment, and control indicates metformin as the first-choice drug, followed by a combination of metformin-sulphonylurea (SU) [7]. The passage of metformin across cell membranes is heavily dependent on its well-known transporters. The most common variant of *SLC22A2* (OCT2) is involved in cDNA change from G to T in position 808 (G808T), which provokes a change from alanine to serine in codon 270 (Ala270Ser) rs316019. This *SLC22A2* variant has been significantly associated with metformin renal clearance among healthy Chinese patients (18.7% vs. 48.2%, *p* = 0.029) [8]. Chen et al. [9] studied metformin’s effect on glucose levels using mice, where variant rs2076828 in *SLC22A3* (OCT3) was associated with a reduced metformin response, because transporters carrying the G allele presented a more pronounced glucose decrease (*p* < 0.05).

Seitz et al. [10] studied 53 populations worldwide and reported that most individuals from Asia and Oceania present fully functional organic cation transporter 1 (OCT1), whereas up to 80% of Latin American ethnicities carry mutations involving OCT1 function loss (major and minor). They also characterized in vitro OCT1 variants on which 10 of its transported substrates were used, including metformin. Mato et al. [11] reported the assessment of rs72552763 (1258_1260del ATG) from the gene-encoding OCT1, *SLC22A1*, across five studies and its frequency variation between 18 and 28%. This report included the association of rs622342 (1386 A > C) with metformin response among three populations (South Indian, Danish, and Caucasian), with a variation of between 5 and 37%. There was no association with HbA1c levels in the Danish population, nor in the Caucasian groups [11]. A study on Caucasian individuals [12] identified a positive association between rs622342 and HbA1c levels (*p* = 0.005) where an average reduction of 0.28% was observed. In contrast, the major A allele of rs622342 from *SLC22A1* displayed a larger probability of responding to metformin treatment in a South Indian population [13].

*ABCB1*, also known as multi-drug resistance protein 1 (MDRP1), is one of 49 putative members in the superfamily of human adenosine triphosphate (ATP)-binding cassette (ABC) transporters, which encode transporter and channel proteins that function as efflux pumps. Hemauer [14] determined the role of some transporters in the bioavailability of oral hypoglycemiants using placental brush border membrane inside- out vesicles (IOVs) and metformin marked with ^14^C- (specific activity, 50 mCi/mmol). Apparent Kt and Vmax values in metformin transport regulated by P-glycoprotein 1 (P-gp1) (100 ± 85 nM and 34 ± 10 pmol/mg protein X minute) highlighted metformin as a high-affinity substrate of placental P-gp used in both monotherapy and combined treatment of gestational diabetes.

Hyung Gyun Kim et al. [15] reported that metformin significantly inhibits the expression of MDR1 by blocking its transcription as it also significantly increases the intracellular accumulation of the fluorescent substrate P-gp-rhodamine-123. Metformin treatment reduces the activity of nuclear factor-kB (NF-kB) and the degradation level of Ikappa B kinase (IkB). Methods employing small interference RNA confirm that a reduction in AMP-activated protein kinase (AMPK) levels mitigates the inhibition of MDR1 activation associated with metformin exposure. AMPK plays an important role in regulating the metformin-induced expression of MDR1.

We recently described the frequencies and clinical implications of *CYP2C9* polymorphisms in DMT2 Mexican-Mestizo patients from Mexico City and we also compared them with reported frequencies from different worldwide analogue populations [16]. Menjívar et al. [17] also reported the polymorphism frequencies of this gene in Mexico City. The majority of Mexican patients with diabetes mellitus type 2 (DMT2) (67.9–85.0%) are prescribed sulphonylureas (SUs), which are metabolized by cytochrome P450 *2C9* (abbreviated as *CYP2C9*). SUs are a type of oral anti-diabetic compound that inhibit ATP-sensitive potassium channels, thus inducing glucose-independent insulin release by the β-pancreatic cells. The wide variability reported in SU responses has been attributed to the polymorphisms of *CYP2C9*. The frequency of *CYP2C9*1/*2* is 50% lower among patients with DMT2 compared with healthy individuals [16]. Considering the widely heterogeneous ethnic background of Mexican populations (which comprise > 65 groups) [18], it is important to further persist in studying the pharmacogenetics of its patients, widening our studies towards OCT1, OCT2, OCT3, and other transporters like *ABCB1* that may be involved, as well as polymorphisms of glibenclamide metabolizing enzymes (*CYP2C9*) in view of its widespread prescription in Mexico.

The aim of the present study is to determine the possible association of polymorphisms OCT1, OCT2, OCT3, *ABCB1*, and *CYP2C9* with HbA1c glycemic control within a sample of DMT2 Mexican-Mestizo patients undergoing metformin or metformin + glibenclamide treatment, as well as identifying those factors associated with glycemic control failure.

## 2. Results

### General Traits of Patient Groups According to Treatment

Biochemical and anthropometric parameters were compared across 103 patients grouped according to treatment type (metformin/metformin + glibenclamide). Statistical significance was found regarding weight, treatment type, and BMI. Fasting glucose levels were 115.5 (102–157) (mg/100 mL) in the monotherapy group and 192.5 (139–238.5) (mg/100 mL) in the combined therapy group (*p* < 0.001). HbA1c levels were 6.4 (5.9–7.3) vs. 9.05 (7.5–10.9) (*p* < 0.001), respectively, for monotherapy and combined therapy. Control proportion defined by %HbA1c was 72.9% in the monotherapy group and 11.4% in combined therapy (*p* < 0.001). This result may be ascribed to the disease’s shorter evolution period, as was reported in the patient follow-up, which indicates that residual insulin secretory capacity, having decreased by 50% at the time of diagnosis, had further decreases of 15% six years later [19]. Metformin dosage in the monotherapy group was 1700 (850–1700) mg/day and 2125 (1700–2550) mg/day in combined therapy (*p* = 0.001). Metformin dosage expressed in terms of mg/kg/day was 19.25 (11.9–25.8) vs. 29.2 (21.5–34.2) (*p* < 0.001), respectively, when comparing monotherapy against combined therapy (Table 1).

We obtained allelic and genotypic frequencies of rs72552763 and rs622342 in OCT1 (*SLC22A1*); rs316019 in OCT2 (*SLC22A2*); rs2076828 in OCT3 (*SLC22A3*); and rs1128503, rs1045642, and rs2032582 in *ABCB1*, and alleles **1, *2* rs1799853, **3,* and *CYP2C9*6* were unsuccessfully tracked within the 204 available samples of Mexican patients with DMT2. Allele **6* is a deletion of a pair of bases in position 818, which produces a premature stop codon and codifies an inactive truncate protein. Its frequency is low among the majority of populations [20] carrying with rs1057910 and rs1934969 (*IVS8-109A > T*) in *CYP2C9*. Every analyzed polymorphism was in Hardy–Weinberg equilibrium (Table 2). We also found no significant differences when comparing frequencies of rs1128503, rs1045642, and rs2032582 in *ABCB1* from our sample with data from a healthy Mexican-Mestizo sample reported by Ortega [21]. When we compared the observed frequencies from our sample with data from the 1000 Genome Project (Clarke) [22], we only found significant differences with respect to African and Asian populations in variants rs2032582 and rs1045642, whereas variant rs1128503 showed frequency differences with respect to most reported populations (unpublished).

As shown in Table 3, genotypic frequencies of the studied polymorphisms were no different when comparing controlled (HbA1c < 7%) and uncontrolled patients (HbA1c ≥ 7%).

We analyzed metformin plasmatic concentrations according to control (defined by %HbA1c) and genotype (Table 4). We observed no significance regarding rs72552763 and rs622342 in OCT1, rs316019 in OCT2, or rs2076828 in OCT3 (Figure 1)

Higher metformin concentrations were observed among non-controlled patients carrying rs1128503 in C/T (*p* = 0.011), rs2032582 in G/T (*p* = 0.041), and rs1045642 in C/T (*p* = 0.002). Among controlled patients, we found the lowest metformin concentration by rs1045642 in C/T (*p* = 0.014) (Table 4) (Figure 2).

The comparison of rs 1799853, rs1057910, and rs1934969 in CYP2C9, showed no statistical significance in plasmatic concentrations across either intra-group or extra-group treatments (Figure 3).

On the other hand, logistic regressions accounting for adjusted dose in terms of kg/weight/day and metformin plasmatic concentration across the three polymorphisms in *ABCB1* found a correlation only with *C/T* of rs1045642 (*p* = 0.017) (Figure 4).

Table 5 shows that patients undergoing combined treatment presented significantly higher %HbA1c (*p* < 0.05), except for OCT1 (rs622342) in C/C, OCT2 (rs316019) in A/C, OCT3 (rs2076828) in G/G, *ABCB1* (rs2032582) in G/T, and G/A, *CYP2C9* *1/*2, *CP2C9* *1/*3, and *CP2C9 IVS-8* in T/T. In the monotherapy group, OCT1 (rs72552763) in GAT/GAT reported a significantly lower %HbA1c (*p* = 0.022) with respect to other genotypes. Significantly higher HbA1c levels were found by OCT1 (rs622342) in C/C (*p* = 0.009), as well as *ABCB1* polymorphisms rs1128503 in C/T (*p* = 0.021) and rs2032582 in G/T (*p* = 0.009).

Each polymorphism was examined by way of three multiple logistic regression models that revealed statistical significances when HbA1c levels were evaluated across genotypes: rs72552763 and rs622342 in OCT1, and rs1128503 and rs2032582 in *ABCB1*. There was no significant association in codominance models regarding the evaluated polymorphisms (Table 6). Moreover, there was no association with no-control according to the simple model adjusted by age, treatment period, treatment type, and BMI apropos of any studied polymorphism, as evidenced by the forest plots (Appendix A).

Two multiple logistic regression models surveyed diplotypes in OCT1 and *ABCB1*, displaying statistically significant covariates. Through the first model, we found significance in age (OR = 0.936 (0.883–0.992); *p* = 0.026), treatment period (OR = 1.261 (1.094–1.452); *p* = 0.001), BMI ≥ 25 kg/m^2^ (OR = 7.049 (1.059–46.895); *p* = 0.043), and combined therapy (OR = 18.05 (5.015–64.969); *p* < 0.001). There was no association with diplotype *GAT/GAT* of rs72552763 or *A/A* rs622342, both in OCT1. The second model revealed significance in age (OR = 0.930 (0.877–0.987); *p* = 0.017), treatment period (OR = 1.271 (1.105–1.462); *p* = 0.001), BMI ≥ 25 kg/m^2^ (OR = 9.128 (1.083–76.966); *p* = 0.042), and combined therapy (OR = 17.933 (4.858–66.205); *p* < 0.001). In this model, there was no association with diplotype *C/C* of rs1128503 or *G/G* of rs2032582, both in *ABCB1* (Table 7).

When comparing dose/mg/kg (Table 8) across *C/T* of rs1128503 in *ABCB1*, we found statistical differences in combined therapy patients as opposed to monotherapy (*p* = 0.001), as well as in *G/G* of rs2032582 (*p* = 0.011) and *C/C* and *T/T* of rs1045642 (*p* = 0.001 and 0.019, respectively). However, metformin plasmatic concentrations across these genotypes presented significant differences only by *C/T* of rs1128503 in *ABCB1*, where they were higher in the combined therapy group (*p* = 0.002) (Figure 5).

## 3. Discussion

We have reported anthropometric data and biomarkers of a Mexican-Mestizo DMT2 patient sample, which was predominantly female (70.8%) with an average age of 54.45 years, and the treatment period percentage, as reported in Table 1, was 60% lower in the monotherapy group with respect to combined therapy patients. Such results were previously reported in Kahan’s seminal studies in a randomized multicenter, double-blind controlled clinical trial called “A Diabetes Outcome Progression Trial” (ADOPT), designed to evaluate the durability of glycemic control in patients undergoing monotherapy with different hypoglycemiants [23], as well as in AL-Eitan’s work [24]. According to its HbA1c markers, the studied sample we hereby discuss presented low control rates (53.39%) and no renal damage according to its glomerular filtration rate (MDRD-4). There were statistical differences between the prescribed treatments regarding dosage mg/day (1700 (850–1700) in monotherapy vs. 2125 (1700–2550) in combined therapy, *p* = 0.001) and dose/weight adjustment in terms of mg/kg/day (19.25 (11.92–25.84) vs. 29.2 (21.55–34.21) (*p* < 0.001)). Therefore, we analyzed whether this was reflected in metformin plasmatic concentrations. Every studied polymorphism within the population of this sample was found in Hardy–Weinberg equilibrium. This particular study produced data on the three polymorphisms in *ABCB1* and its genotypes among a population of Mexican DMT2 whose frequencies were previously unknown; the analysis found no statistical difference across genotypic frequencies between these populations and healthy Mexican-Mestizo volunteers [21].

When sorting drug plasmatic concentrations according to treatment type and genotype, we found a statistically significant difference only by *C/T* of rs1128503 in *ABCB1* among the combined therapy group. Metformin is transported into hepatocytes principally by the product of *SLC22A1* (OCT1), as well as *SLC22A2* (OCT2) and *SLC22A3* (OCT3). It is also a substrate of all three polymorphic products of *ABCB1* (P-gp 1); rs1128503, rs2032582, and rs1045642 [14]. All of these genotypes are likely to impact metformin’s pharmacokinetics. Ruicheng [25] found no association between the genotypes of rs1128503 in *ABCB1* and diabetes risk in the Chinese Han population, adducing its intronic nature and introns’ characteristic function loss during evolution. Llaudó [26] described the effect of Pgp activity decrease in PMBC through Rho123 efflux among patients with transplanted kidneys receiving immunosuppressants, who were carriers of allele T/T of rs1045642, and therefore called them “low pumpers,” as opposed to *C/C* and *C/T* of rs1045642, the “high pumpers”. As for rs72552763 in OCT1, Menjivar [17] reported an association with no-control in a Mexican population, specifically, with *GAT/GAT*, who amounted to 66% of uncontrolled patients (*p* = 0.011). In our study, no-control frequency was 38.2% by *GAT/GAT* of rs72552763, which was even higher by *del/GAT* at 50.9%, although these results accounted for no statistical difference (*p* = 0.392). We attribute these dissimilarities to our country’s ethnic diversity. In the Rotterdam study, which included Caucasian patients above 55 years of age undergoing occasional metformin intake, Becker [12] reported the decreasing effect on HbA1c levels by rs2289669, which codifies efflux transporter MATE1; these patients also carried *CC* of rs622342, and they presented a significant association with HbA1c changes (−0.68; 95% CI: −1.06 to −0.30; *p* = 0.005). Abdel-Hameed [27] reported that variant rs622342 of *SLC22A1*(OCT1) is associated with the therapeutic response to the metformin + glibenclamide combined treatment, where allele *AA* responded 2.7 times more to metformin with respect to allele *C*, as shown in a recessive model (recessive model, odds ratio 2.718, *p* = 0.025, 95% CI 1.112–6.385). In that particular study, disease duration reported was an average of 2.6 years among non-responders and 2.3 years in the responder group. More recently, in a Mexican sample, Resendiz-Abarca [28] identified genotypes *CC*-rs622342, *AA*-rs62803, and *GG*-rs594709, associated with increased HbA1c levels after a 12-month metformin treatment. In her study on a Mexican-Mestizo population, Menjivar [17] found no association between no-control and any genotypes of rs622342 in OCT1 (*p* = 0.066). Our study coincides with these data since we did not observe an association between no-control and rs622342 either (*p* = 0.324). We found no significant differences through the analysis of either control or no-control frequencies regarding any polymorphism of those genes expressing OCT1 (rs72552763 and rs622342), OCT2 (rs316019), OCT3 (rs2076828), *ABCB1* (rs1128503; rs2032582; rs1045642), or *CYP2C9* (rs1799853; rs1057910; rs1934969). After sorting each polymorphism according to HbA1c levels and treatment type, we found several statistical differences across every genotype of rs72552763, rs622342 (except for *C/C*), *C/C* of rs316019, *C/C* and *C/G* of rs2076828; every genotype of rs1128503 in *ABCB1*, and *G/G* and *T/T* of rs2032582; every genotype of rs1045642; the genotype **1/*1* of *CYP2C9*; and *A/A* and *A/T* of rs1934969. This proves that combined therapy does not reduce HbA1c levels, but rather quite the opposite—it preserves higher levels with respect to monotherapy. This may also be ascribable to treatment period differences. Kahan [23] reported the occurrence of metformin monotherapy failure within 5 years in 21% of cases; in our sample, combined therapy patients already reported 7.5 years with the disease. In logistic regression models performed on HbA1c no-control, where the dominant genotypic model was adjusted by age, treatment period, treatment type, and BMI, we found no association with no-control in genotypes of OCT1 (rs72552763), OCT1 (rs622342), *ABCB1* (rs1128503), and *ABCB1* (rs2032582), as shown by an adjusted simple/multiple logistic regression model adjusted in accordance with the aforementioned covariates.

The literature is contrasting regarding possible associations between polymorphisms and HbA1c levels. Ethnicity, treatment type, and treatment period affect results heterogeneously, whereas clinical trial design is complicated compared to observational studies such as ours. At this point, it would seem reckless to advance a conclusive statement about our results. Further analysis of metformin plasmatic concentrations is necessary, and if possible, associations with the chosen polymorphisms are to be determined. By researching the probable association between these and the HbA1c control, our study marks the first evaluation of metformin plasmatic concentrations among DMT2 Mexican-Mestizo carriers of these particular polymorphisms, whether they were undergoing metformin monotherapy or the metformin + glibenclamide combination (Table 4). P-gp’s probable role in this disease was not considered either, although metformin is a known substrate of this efflux pump. We are not aware of any other similar study. For this analysis of *ABCB1* polymorphisms, we grouped patients by HbA1c percentage, whether they were controlled or not. In *C/T* of rs1128503, patients with HbA1c < 7% presented metformin plasmatic concentrations (ng/mL) lower than HbA1c ≥ 7% (139.8 (90.3–649.7) vs. 844.7 (481.7–1095.1); *p* = 0.011). This genotype received a higher metformin + glibenclamide dose (mg/kg/day) than metformin monotherapy (*p* = 0.011), which may explain the observed effect. Likewise, patients carrying *G/T* of rs2032582 with HbA1c < 7% reported lower metformin plasmatic concentrations (ng/mL) than HbA1c ≥ 7% 181.1 (118.2–768.5) vs. 491.4 (185.9–884.2) (*p* = 0.041). However, this genotype did not receive a higher metformin + glibenclamide dose than monotherapy (*p* = 0.170); hence, we consider that this therapeutic effect cannot be ascribed to a wider metformin bioavailability. In *C/T* of rs1045642, patients with HbA1c < 7% reported lower metformin plasmatic concentrations (ng/mL) than HbA1c ≥ 7% [116.8 (85.7–157.8)] vs. (530.7 (185.9–1095.1); *p* = 0.002). In this last polymorphism of rs1045642, we also observed statistical differences within groups across those genotypes in monotherapy treatment (*p* = 0.014). This last group undergoing combined therapy did not receive higher dosages mg/kg/day than monotherapy (*p* = 0.152), just like *G/T* of rs2032582. The therapeutic effect cannot be ascribed to a larger metformin bioavailability, but it could be related to P-gp 1 high pumpers, as described by Llaudó [26]. Recently, AL-Eitan [24] genotyped 21 polymorphisms of OCT1, OCT2, and OCT3 among DMT2 patients from Northern Jordan: seven polymorphisms of OCT1, 10 of OCT2, and four of OCT3. AL-Eitan reported no statistical difference regarding glycemic control OCT (according to the ADA’s criteria). Their conclusions reported that rs12194182 in *SLC22A3* (OCT3) was associated with better HbA1c levels. Genotypes of the studied SNPs in *SLC22A1, SLC22A2*, and *SLC22A3* were significantly associated with BMI, age at the time of diagnosis, and glycemic control as established through multinomial logistic regression (*p* < 0.05). Treatment period was not considered. Finally, there was no relation between SNPs and glycemic control after age adjustment (*p* > 0.05).

We performed a logistic regression analysis using a dominant genotypic model including rs72552763 and rs622342 in OCT1, as well as rs1128503 and rs2032582 in *ABCB1*. Since we found no significant difference, we conducted a second multiple logistic regression analysis to confirm whether rs72552763 *GAT/GAT* and rs622342 *A/A* in *OCT1*, as well as rs1128503 *C/C* and rs2032582 *G/G* in *ABCB1*, showed a significant glycemic control *p*-value; we found none, unlike about other covariates such as age, BMI, combined treatment, and treatment period. All of the latter reported statistical significance in both models, where the highest no-control ORs were OR = 18.05 by OCT1 and OR = 17.93 by *ABCB1* in combined therapy. This became patent in the forest plot we present (Appendix A). To summarize, none of the polymorphisms we studied reported glycemic control association with any of their genotypes, as opposed to some covariates that were indeed associated with no-control. In our sample, low therapeutic efficacy cannot be attributed to metformin but rather to its covariates. The evidence of such covariates’ significant role in uncontrolled HbA1c levels (the golden standard biomarker) may benefit Mexican DMT2 patients in the form of a personalized prescription, favoring a more adequate therapeutic intervention and lifestyle changes in order to prevent micro and macrovascular complications hauled by the disease.

## 4. Materials and Methods

### 4.1. Study Design and Sample Description (DMT2 Patients)

This observational study recruited participants previously diagnosed with DMT2 according to the World Health Organization and American Diabetes Association standards (ADA) [29]. All of them were legal adults (18 years or older) with at least three generations of Mexican ancestry (assessed through a questionnaire) and no kinship between them. Sample collection and clinical record reviewing was accomplished within a cohort of 103 patients with DMT2 undergoing medical treatment and monitored at a third-level public healthcare center in Ixtapaluca, Mexico, between May 2018 and December 2019. All participants provided written informed consent. The research protocol was ethically approved by a Research and Ethics Commission of the National Autonomous University of Mexico (Faculty of Medicine) 001/SR/2016, and the study was performed in accordance with the Declaration of Helsinki and the 64th WMA General Assembly, Fortaleza, Brazil, October 2013, on ethical principles for medical research involving human subjects. A database of the collected clinical and biochemical data, including each patient’s file, was compiled. Files were thoroughly reviewed in accordance with the study’s inclusion and exclusion criteria. Patients were subsequently grouped in accordance with the designated hypoglycemic agent used for treatment: metformin monotherapy and glibenclamide + metformin.

Inclusion criteria required at least 3 months of pharmacological treatment in order to report stationary plasmatic levels. HbA1c values were used to estimate plasmatic glucose levels during the whole trimester prior to their determination. The present study focused on a third-level center where patients’ files spanning anywhere from a semester to multiple years could be revised. Determining previous measurements would not have been achievable and the present study was only interested in verifying control over the last trimester.

### 4.2. Clinical Evaluation

Out of the 204 initially evaluated profiles, 52 were eliminated due to not meeting inclusion criteria. Of these remaining 152 allocated for observation, 26 had no HbA1c record and 23 had no plasmatic metformin record; thus, only 103 were fit for biomarker analysis (Figure 1).

Patients were recruited according to the following inclusion criteria: (i) The patient was undergoing either metformin treatment, or a combination of metformin + glibenclamide; (ii) the patient had undergone a treatment schedule comprising a stable dose of these drugs for at least 3 months; (iii) the precedents and treatment characteristics of each individual were accessible via their medical record at the corresponding healthcare center, particularly data concerning drug dosage (including hypoglycemic agents) during the aforementioned 3-month period; (iv) the medical file comprised anthropometric parameters [30] and clinical laboratory reports performed at the Hospital Regional on a number of key biochemical variables (including HbA1c through HPLC, in a Variant II Turbo 2.0, Bio-Rad, Hercules CA, USA; fasting glucose levels, total cholesterol, LDL, HDL, triglycerides, creatinine by photometry in an AU480 Chemistry Analyzer, Beckman Coulter, Brea CA, USA). Individuals who reported chronic alcoholism, previous pancreatic pathology, renal failure, hypoglycemic treatment with insulin or insulin analogs, insufficient medical records, DMT1, or voluntary withdrawal were excluded. A database was created to retrieve and analyze the information on the 103 patients included in the study. File revision was performed through random probabilistic sampling.

### 4.3. Genotyping Procedure

A peripheral 10 mL blood sample was collected from all participants in EDTA tubes, and genomic DNA was extracted from 200 μL of each patient’s venous peripheral blood using UltraClean^®^ BloodSpin^®^ DNA isolation reagents (Mo Bio Laboratories; Qiagen, Inc., Valencia, CA, USA), evaluated for integrity and concentration through 1% agarose electrophoresis and spectrophotometry using NanoDrop™ 2000/2000c (Thermo Scientific, Wilmington DE, USA), respectively. For *CYP2C9, SLC22A1, SLC22A2, SLC22A3*, and *ABCB1* allele determinations, different allelic variants were analyzed by RT-PCR technology using fluorescence-based TaqMan^®^ assays on a Fast 7300 Real-Time PCR System, both from Applied Biosystems (Thermo Fisher Scientific, Foster City, CA, USA). Reactions were performed in a final reaction volume of 10 μL, with 30 ng genomic DNA template, 1X TaqMan^®^ Universal PCR Master mix system from Applied Biosystems, 1X each probe assessed (*CYP2C9*1*: C_160889442_10, *CYP2C9*2*: rs1799853, C_25625805_10; *CYP2C9*3*: rs1057910, C_27104892_10; CYP2C9*6: rs9332131, C_32287221_20; SLC22A1: rs12208357, C__30634096_10; SLC22A1: rs2282143, C__15877554_40; SLC22A1: rs594709, C___1898206_20; SLC22A1: rs622342, C____928527_30; SLC22A1: rs628031, C___8709275_60; SLC22A1: rs683369, C____928536_30; SLC22A1: rs72552763, C__34211613_10; SLC22A2: rs316019, C___3111809_20; SLC22A3: rs2076828, C___2763995_1_; SLC22A3: rs8187725, C__30633894_10; ABCB1: rs1045642, C___7586657_20; ABCB1: rs1128503, C___7586662_10; ABCB1: rs2032582, C_11711720D_40; ABCB1: rs2032582, C_11711720C_30; ABCB1: rs2032588, C__11711718_10), and water. Thermocycling conditions and allelic discrimination to identify the genotypes using allelic discrimination software (Applied Biosystems, Bedford, MA, USA) were previously described [16]. The intronic polymorphism *CYP2C9 IVS8 109A > T* (rs1934969) was analyzed using the PCR amplifying enzymatic restriction fragment long polymorphism (PCR/RFLP) method. PCR was performed on a Mastercycler^®^ 384 (Eppendorf, Hamburg, Germany) to identify its presence by following the procedure detailed in Cuautle et al. (2019) [16]. Fragments of 468 and 154 bp cut by HinfI restriction enzyme (New England BioLabs Inc., Ipswitch, MA, USA; cat no. RO155S) were observed for *CYP2C9 IVS8 109T*, whereas a band of 622bp corresponding to uncut fragments was matched to the *CYP2C9 IVS8 109A > T* allele. SNP allelic and genotypic frequencies of OCT1, OCT2, OCT3, *ABCB1*, and *CYP2C9* were carried out through direct counting.

### 4.4. Plasmatic Metformin Determination

Out of the initial 204 candidates, 101 were excluded from the analyses for the following reasons: 2 plasmatic samples were insufficient, 14 were reported as unquantified, and 29 were reported as undetermined (total 45), whereas 56 more were lost during follow-up (41 changed treatment and 15 had no HbA1c record) (Figure 1). Out of the remaining 103 included patients, plasmatic concentrations could only be determined in 86, since 9 had been registered as undetermined and 8 more were not quantified (either hemolyzed or measured through lipemic serum). Determinations were carried out in the Clinical Pharmacology Unit of UNAM’s Faculty of Medicine. The methodology was validated in accordance with the Mexican Official Normativity NOM-177-SSA 1-2013 [31], which establishes tests and procedures to demonstrate a drug’s interchangeability; the mandatory requirements authorized third parties must observe; which research or healthcare institutions may perform biocompatibility tests; and the internal procedure of analytical methodology validation. The study also adhered to additional international requirements, whose acceptance parameters were established in the Standard Operating Procedure SOP-UA-05-09 “Validation of analytical methodology on special and bioavailability and/or bioequivalence studies.” To analyze biological samples, we employed the method described in the analytical methodology index card FMA-018/B, which had been previously validated according to Mexican Official Normativity NOM-177-SSA1-2013. The analytical method was selective over the quantification of both plasmatic metformin and glibenclamide, without the interference of either endogenous or exogenous compounds. The employed methodology proved to be selective, linear, precise, and exact over the assessed concentration range.

For sample analysis, we employed UHPLC-MS/MS in MRM mode using an Agilent Technologies G6490A mass spectrometer. In the preparation of calibration curves and controls for sample analysis we employed the following reference substances: metformin hydrochloride (U.S.P. batch R069H0, purity 99.7%), glibenclamide (U.S.P. batch R022S0, purity 99.4%), and loratadine (U.S.P. R052U0, purity 99.8%). To quantify plasmatic metformin/glibenclamide, we employed the mass/charge ratio of metformin 130.1/71.0, glibenclamide 494.0/369.0, and the internal standard loratadine 383.1/337.1. A Luna PFP analytical column (2.0X100 mm, 3.0 µm) from Phenomenex was used for analyte separation and determination. The isocratic elution of samples was carried out by using acidified ammonium formate 10 mM (A): acetonitrile 100% (B), as mobile phase. Analytes were previously extracted through protein extraction/precipitation: A 100 μL aliquot was extracted from a plasma sample and subsequently deposited into a microtube. We added a 10 μL aliquot from the loratadine internal standard solution (30 μg/mL). To carry out protein precipitation, we added a 400 μL aliquot of HPLC-grade acetonitrile. The tube was shaken on a multiple vortex at maximum speed for 1 min. The tube was centrifuged at 13,000 rpm and 4 °C for 5 min. We recovered 250 μL of the supernatant and transferred it to a 96-well plate. The injection volume on the chromatographic system was 2.0 µL. The method was linear in the range of 20–10,000 ng/mL. Intra-day and inter-day variation coefficients were less than 15%. In the case of metformin, recovery ranged from 89.676 to 90.731%. The relationship between chromatographic response and concentration on every calibration curve was adjusted through linear least squares regression for metformin. To quantify the plasmatic samples, the regression was performed through Mass Hunter B.08 Quantitative Analysis software.

Patients were summoned by their respective treating physician having observed a fasting period of at least 8 h. All of the blood samples were taken within an interval of 8 h after the evening’s metformin dose. A 10 mL peripheral venous blood sample was extracted using EDTA vacutainer tubes. The sample was centrifuged at 400× *g* for 5 min at 4 °C. Once the plasma was obtained, aliquots were carried out using Eppendorf tubes and the samples were frozen at −80 °C until drug determination assays were simultaneously performed across all of them.

### 4.5. Statistical Analyses

Data distribution was assessed through Kolmogorov–Smirnov and Shapiro–Wilk tests. Normal distribution variables are displayed as median and standard deviation, whereas free distribution variables are displayed as median and interquartile ranges. Patients were grouped according to (i) HbA1c control, no-control; (ii) treatment type; and (iii) polymorphisms of *SLC22A1*(OCT1), *SLC22A2*(OCT2), *SLC22A3*(OCT3), *ABCB1*, and *CYP2C9*, where %HbA1c and metformin plasmatic concentration were evaluated. Normal distribution quantitative variables were compared through Student’s *t*-test, whereas free distribution variables were compared using Mann–Whitney’s U-test. When comparing free distribution quantitative variables across 3 groups, the Kruskal–Wallis test was employed. Qualitative variables were compared using Pearson’s Chi square test or Fisher’s exact test if necessary. Ordinal variables were compared through the Mann–Whitney U-test. A *p* value < 0.05 was considered statistically significant. Analyses were performed using SPSS version 23.0 for Windows (IBM Corp., Armonk, NY, USA).

### 4.6. Genotypic and Allelic Frequency Analysis

Allelic frequencies were counted, and respective expected values were calculated for each genotype. A *p* value > 0.05 defined the frequencies in Hardy–Weinberg equilibrium as calculated through Pearson’s Chi square test.

### 4.7. Logistic Regression

We focused on statistically significant HbA1c SNPs, whose genotypes were rs72552763 and rs622342 in OCT1, and rs1128503 and 2032582 in *ABCB1* (Table 6). A codominance model was elaborated for each SNP, assuming every genotype implied a different risk for the dependent variable, which was no-control defined as HbA1c ≥ 7%. Additionally, each SNP had its own simple and multiple logistic regression model observing genotypic dominance. The multiple model was adjusted according to age, treatment period, treatment type, and BMI, which showed OR (IC95%) in every case. Statistical significance was *p*-value < 0.05.

A second multiple logistic regression model was conducted considering rs72552763 and rs622342 in OCT1, and rs1128503 and 2032582 in *ABCB1* as diplotypes. Reference values for qualitative variables were BMI under 25 kg/m^2^, metformin therapy, and both heterozygous and recessive genotype in every model of the analyzed polymorphisms. The dependent variable was no-control defined as HbA1c ≥ 7%. References were selected this way to explore no-control risk for a patient with a BMI over 25 kg/m^2^ undergoing combined therapy and carrying genotypes rs72552763 *GAT/GAT* with rs622342 *A/A* in a first model and rs1128503 *C/C* with rs2032582 *G/G* in a second model.

## 5. Conclusions

This study’s results seem adequate to become part of a predictive algorithm for glycemic control, considering variables like age, BMI, treatment, and disease period. This would be quite helpful in clinical interventions specifically aimed towards populations mathematically predisposed to non-control risk. Such an instrument could diminish comorbidities and complications associated with diabetes. Study limitations: This is essentially an observational study based on a small sample. We were able to quantify metformin plasmatic concentrations only once. Patient records were often missing data, and it was quite difficult to properly follow up with them.

## Data Availability

The data presented in this study are available on request via the corresponding author. These data are not publicly available because the patients and researchers are bound to an agreement establishing that only the head of the study and Mexican health authorities shall have access to them, in accordance with the presidential decree of 16 April 2015, sanctioning the General Law on Transparency and Access to Public Information.

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
