# Peer review of "Pharmacogenetics of Metformin Transporters Suggests No Association with Therapeutic Inefficacy among Diabetes Type 2 Mexican Patients"

_pharmaceuticals, 2022, doi:10.3390/ph15070774_

Round 1

Reviewer 1 Report

The manuscript has been sufficiently redrafted

Author Response

A kind note: R1 (Accept) is the previous R3 (Major)

Reviewer 2 Report

The authors have improved the paper in many ways.

The abstract do not reflect the possible associations found in the comparisons performed between genotypes, overall in HbA1c and do not explain limitations of the study (overall the important differences in the treatment groups).

Reviewer 3 Report

This study presents interesting results on pharmacogenetics of metformin transporters in Mexican patients with type 2 diabetes. Each step of the study including analysis of metformin concentrations, genetic tests and statistical analyses seem to be appropriately performed. The findings are clinically important but the manuscript should be better organized. The study has several issues that need to be addressed:

1.     Page 2, lines 55-57: the sentence is unclear

2.     Page 2, line 62: “both functional alleles” – Authors should specify which alleles are functional

3.     Page 2, line 81: the “P-gp” abbreviation should be explained when used for the first time

4.     Page 2, line 91: the “IkB” abbreviation should be explained

5.     Page 2, lines 91 and 93: the “AMPK”  abbreviation is explained twice

6.     Page 2-3, lines 95-102: this part does not refer to the studied polymorphisms and should be deleted or relocated

7.     Page 3, lines 105-109: what is the main conclusion regarding CYP2C9 polymorhisms? Authors should provide rationales for studying particular CYP2C9 variants

8.     Page 3, lines 116-119: should be deleted as the same message was provided in lines 121-125.

9.     Page 3, lines 135-137: Do Authors have any explanation for better control proportion of %HbA1c when metformin was given in monotherapy than in combined therapy?

10.  Page 3, lines145: Authors state that *1, *2, and *3 CYP2C9 alleles were studied but on page 20, the PCR conditions for CYP2C9*2, *3 and *6 were provided

11.  Table 2 and 3 could be moved to supplementary file 

12.  Titles of tables 3 and 4 are exactly the same although they present different results

13.  Page 7, lines 177-192: Authors repeat values that are provided in table 4. More critical evaluation of the meaning of the results would be more interesting for readers.

14.  Figure 1, 2 and 3 present the same results that are provided in table 5. Additionally, in lines 199-211 the values from the table were repeated. Authors should move table 5 to supplementary file or modify the description

15.  Page 11, line 241-256:  repetition of the values from table 6. Authors should present summary of the results without numerical values which are clearly presented in table 6

16.  Figure 5 contains exactly the same information as table 8 and provided in lines 296-307. Figure 5 should be removed

17.  Page 16, line 327: please revise your data regarding percentage of women. According to table 1 they constitute > 70% of the studied population

18.  Page 16, line 328: what do Authors mean stating that “the disease period was 40% inferior in the monotherapy group”?

19.  Line 341: provide reference

20.  4.1. Study design: there is no information on the local bioethics committee agreement 

21.  Line 536: provide reference to the PCR procedure used in the study

22.  Lines 566-582: provide accuracy and precision of the HPLC-MS/MS method, which agents were used for protein precipitation?

23.  Lines 584-589: what was the sampling time elapsed form metformin administration?

24.  Line 628: “HbA1c SNPs” is not clear for the readers

Author Response

This study presents interesting results on pharmacogenetics of metformin transporters in Mexican patients with type 2 diabetes. Each step of the study including analysis of metformin concentrations, genetic tests and statistical analyses seem to be appropriately performed. The findings are clinically important but the manuscript should be better organized. The study has several issues that need to be addressed:

  1. Page 2, lines 55-57: the sentence is unclear

We have changed the sentence to “because transporters carrying the G allele presented a more pronounced glucose decrease (P<0.05).”

  1. Page 2, line 62: “both functional alleles” – Authors should specify which alleles are functional?

R: We have adressed the confusion by modifying the text as follows:

…whereas up to 80% of Latin American ethnicities carry mutations involving OCT1 function loss.      

  1. Page 2, line 81: the “P-gp” abbreviation should be explained when used for the first time

R: P-glycoprotein 1.

  1. Page 2, line 91: the “IkB” abbreviation should be explained

R: Ikappa B kinase.

  1. Page 2, lines 91 and 93: the “AMPK”  abbreviation is explained twice

R: We thank the reviewer for the observation. We have eliminated the second notation.

  1. Page 2-3, lines 95-102: this part does not refer to the studied polymorphisms and should be deleted or relocated

R: We have moved the paragraph to the Discussion section.

  1. Page 3, lines 105-109: what is the main conclusion regarding CYP2C9 polymorhisms? Authors should provide rationales for studying particular CYP2C9 variants

R: We have inserted the following text: The majority of Mexican patients with diabetes mellitus type 2 (DMT2) (67.9-85.0%) are prescribed sulphonylureas (SUs), which are metabolized by cytochrome P450 2C9 (abbreviated as CYP2C9). SUs are a type of oral anti-diabetic compound which inhibit ATP-sensitive potassium channels, thus inducing glucose-independent insulin release by the β-pancreatic cells. The wide variability reported in SU responses has been attributed to the polymorphisms of CYP2C9. The frequency of CYP2C9*1/*2 is 50% lower among patients with DMT2 compared with healthy individuals [16].

  1. Page 3, lines 116-119: should be deleted as the same message was provided in lines 121-125.

R: We have eliminated the indicated paragraph.

  1. Page 3, lines 135-137: Do Authors have any explanation for better control proportion of %HbA1c when metformin was given in monotherapy than in combined therapy?

R: We have inserted the following text into line 136: this result may be ascribed to the disease’s shorter evolution period, as it has been reported in the follow-up of patients from the control group of the United Kingdom Prospective Diabetes Study UKPDS (United Kingdom Prospective Diabetes Study), which indicates that residual insulin secretory capacity, yet decreased by 50% at the time of diagnosis, further decreases of 15% six years later Guillausseau (21).

  1. Page 3, lines145: Authors state that *1, *2, and *3 CYP2C9 alleles were studied but on page 20, the PCR conditions for CYP2C9*2, *3 and *6 were provided

     R: We have inserted the following text: CYP2C9*6 was unsuccessfully tracked within the 204 available samples of Mexican patients with DMT2. Allele *6 is a deletion of a pair of bases in position 818, which produces a premature stop codon and codifies an inactive truncate protein. Its frequency is low among the majority of populations (25).

  1. Table 2 and 3 could be moved to supplementary file 

       R: Because of Mexico’s wide ethnic variety (65 different groups), we deem it necessary to preserve these tables within the eventual publication. Table 2 reports some SNP allelic and genotypic frequencies (ABCB1) for the first time ever among DMT2 patients. On the other hand, Table 3 succinctly justifies subsequent analyses of greater complexity.

  1. Titles of tables 3 and 4 are exactly the same although they present different results

     R: We thank the reviewer for this observation. The edition of Table 4 was evidently flawed. We have corrected and highlighted the text.

  1. Page 7, lines 177-192: Authors repeat values that are provided in table 4. More critical evaluation of the meaning of the results would be more interesting for readers.

R: We are grateful for the observation. We have modified the result description as follows:

Higher metformin concentrations were observed among non-controlled patients carrying rs1128503 in C/T (P=0.011), rs2032582 in G/T (P=0.041), and rs1045642 in C/T (P=0.002). Among controlled patients, we found the lowest metformin concentration by rs1045642 in C/T (P=0.014).

  1. Figure 1, 2 and 3 present the same results that are provided in table 5. Additionally, in lines 199-211 the values from the table were repeated. Authors should move table 5 to supplementary file or modify the description

R: We are thankful for the observation. We have moved Table 5 to the supplementary files. We have also eliminated the corresponding text.

  1. Page 11, line 241-256:  repetition of the values from table 6. Authors should present summary of the results without numerical values which are clearly presented in table 6

R: Thank you for noting this. We have changed the text as follows:

Patients undergoing combined treatment presented significantly higher %HbA1c (p<0.05), except for OCT1 (rs622342) in C/C, OCT2 (rs316019) in A/C, OCT3 (rs2076828) in G/G, ABCB1 (rs2032582) in G/T and G/A, CYP2C9 *1/*2, CP2C9 *1/*3, and CP2C9 IVS-8 in T/T. In the monotherapy group, OCT1 (rs72552763) in GAT/GAT reported a significantly lower %HbA1c (p= 0.022) respect to other genotypes. Significantly higher HbA1c levels were found by OCT1 (rs622342) in C/C (p=0.009), as well as ABCB1 polymorphisms rs1128503 in C/T (p= 0.021) and rs2032582 in G/T (p= 0.009).

  1. Figure 5 contains exactly the same information as table 8 and provided in lines 296-307. Figure 5 should be removed

R: Figure 5 was moved to the supplementary files, along with its corresponding text.

  1. Page 16, line 327: please revise your data regarding percentage of women. According to table 1 they constitute > 70% of the studied population

R: We are grateful for the observation. The correct percentage (70.87%) has been inserted.

  1. Page 16, line 328: what do Authors mean stating that “the disease period was 40% inferior in the monotherapy group”?

R: We have modified the text as follows:

“The correct treatment percentage, as reported in Table 1, is 60% lower in the monotherapy group with respect to combined therapy patients.”

  1. Line 341: provide reference

R: We have included the reference to Ortega-Vázquez A [20].

  1. 4.1. Study design: there is no information on the local bioethics committee agreement 

R: The Protocol Registration 001/SR/2016, generated by UNAM’s Faculty of Medicine and approved by the Ixtapaluca Hospital, has been added to lines 463-465 in pages 18-19. The section Institutional Review Board Statements includes the agreement too.

  1. Line 536: provide reference to the PCR procedure used in the study

R: Thank you for observing this. We have modified the text as follows:

“(...) detailed in Cuautle, et al, 2019 [16].”    

  1. Lines 566-582: provide accuracy and precision of the HPLC-MS/MS method, which agents were used for protein precipitation?

R: We have now explained the extraction/precipitation method in tis fashion:

100 mL aliquot was extracted from plasma sample and subsequently deposited into a microtube. We added a 10 mL aliquot from the loratadine internal standard solution (30 mg/mL). To carry out protein precipitation, we added a 400 mL aliquot of acetonitrile HPLC grade. The tube was shaken onto a multiple vortex at maximum speed for 1 minute. The tube was centrifuged at 13,000 rpm and 4ºC for 5 minutes. We recovered 250 mL of the supernatant and transferred into a 96-well microplate. The method was linear in the range of 20-10,000 ng/mL. Intra-day and inter-day variation coefficients were less than 15%. In the case of metformin, recovery ranged from 89.676 to 90.731.

  1. Lines 584-589: what was the sampling time elapsed form metformin administration?

R: The text now reads “All of the blood samples were taken within an interval of 8:00 h after the evening’s metformin dose”.

  1. Line 628: “HbA1c SNPs” is not clear for the readers

R: We have adressed the confusion by modifying the text as follows:

We focused on SNPs reporting statistically significant HbA1c differences (table 6).

Round 2

Reviewer 2 Report

Authors have improved the paper following the recomendations of the reviewers. 

Reviewer 3 Report

The authors have replied to all comments and taken into account all my suggestions. The paper is improved to a large extent and it can be published in present form

This manuscript is a resubmission of an earlier submission. The following is a list of the peer review reports and author responses from that submission.

Round 1

Reviewer 1 Report

Present work describes the results of the analysis of a group of genes previously described as involved in the pharmacogenetics of metformin. The authors do not find relevant associations with the metformin levels or in diabetes-related parameters between the treatment groups (metformin and merformin+glibenclamide). The main association with glycemic non-control are age, treatment duration, BMI, and combined treatment (metformin+glibenclamide).

The study is based in the use of two groups of patients with a limited number of participants. In them, the authors have measured metformin levels in blood, but they do not indicate the procedure for reducing variability in metformin levels: moment of sample extraction, time from medication ingestion, hours of fasting, conservation of samples, etc.

In material and methods the authors indicate as inclusion criteria “The patient was undergoing either glibenclamide or metformin treatment, or a combination of both”; in the rest of the test, they indicate that there are only two option of treatment: metformin or merformin+glibenclamide. This can generate an important alteration in the results: Please indicate the number of patients that only are receiving glibenclamide. Comparisons between patients with different genotypes within the same treatment group appear not to have been made.

In whole text, there are many mistakes (missing words, not using upper cases when correspond, etc.) or they do not indicate in a correct way the information. For example, the authors indicate: “Figure 1 shows metformin’s plasmatic concentrations across 159 fasting patients carrying SLC22A1, SLC22A2 and SLC22A3”

Reviewer 2 Report

Comment #1: the group of patients on metformin monotherapy had a disease duration significantly lower than the group of patients on metformin plus glibenclamide. This fact probably explains the large difference between groups in terms of glycemic control (better in the group of patients on metformin monotherapy) – as previous published in seminal studies on this issue (N Engl J Med 2006; 355:2427-2443).

Reviewer 3 Report

Today, the pathogenesis, symptoms, and treatment of type 2 diabetes (DMT2) are well known. The manuscript ID: pharmaceuticals-1595416 provides additional information on the course of DMT2 in Mexican populations.  However, the article needs redrafting. There is no discussion of the results and specific conclusions. The information in the introduction should be included in the discussion. The introduction itself should introduce you to a clearly defined goal.

The research was carried out on a group of 103 people, it is not necessary to describe the entire 204-person group, this misleads the reader. What the methods of assessing blood biochemical parameters were used? Insert references after citing Chen et al.; Correct punctuation errors